# Interannual Variability of Snowiness and Avalanche Activity in the Ile Alatau Ridge, Northern Tien Shan

**Akhmetkal Medeu, Viktor Blagovechshenskiy \*, Tamara Gulyayeva, Vitaliy Zhdanov** 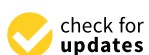 **and Sandugash Ranova**

Institute of Geography and Water Safety, Almaty 050010, Kazakhstan
\* Correspondence: victor.blagov@mail.ru; Tel.: +7-705-768-2378

**Abstract:** Snowiness and avalanche activity are very important natural characteristics of mountain areas. They have a great influence on the possibility of areas' development, especially regarding winter recreation. This article considers the interannual variability of snowiness and avalanche activity in the Ile Alatau Ridge (Northern Tien Shan), which belongs to the areas with a continental snow climate. The sum of winter precipitation and snow depth are used as snowiness indices, and the indices of avalanche activity are the total avalanche volume, maximum avalanche volume and number of avalanches. The work uses archival data for the period from 1966 to 2022. Interannual variability of snowiness and avalanche activity indices and long-term temporal trends were assessed, correlation between these indices was studied, and extreme values with different return periods were calculated. The relationship between years with a high snowiness and years with a high avalanche activity, as well as years with a high avalanche activity and years with a large number of avalanche victims and high avalanche damage has been studied. Similar studies have not been previously carried out for the areas with a continental snow climate. Snowiness indices have weak, non-significant, increasing temporal trends. The total avalanche volume has a non-significant decreasing temporal trend, and the maximum avalanche volume has a significant decreasing one. The number of avalanches has a significant increasing temporal trend. This study could be relevant for understanding the features of temporal variability of snowiness and avalanche activity in the mountainous regions with a continental snow climate.

**Keywords:** avalanche activity; avalanche victims; avalanche volumes; interannual variability; long-term temporal trends; Northern Tien Shan; snow depth; winter snowiness

## 1. Introduction

Snowiness and avalanche activity are very important natural characteristics. They affect the possibilities of using mountainous regions. Snow amount is the main condition for winter sports and recreation development. At the same time, it increases the cost of road maintenance and building operation. A high avalanche activity makes it necessary to spend a lot of money on ensuring avalanche safety. Therefore, snowiness and avalanche activity must be taken into account when planning the development of mountain territories. Not only average long-term indices, but also their interannual variability and temporal trends should be considered. To solve this problem, it is necessary to use a statistical analysis of long-term data on the characteristics of snow cover and avalanche activity.

The purpose of this work is to determine interannual variability of snowiness and avalanche activity in the Ile Alatau Ridge. The significance of the work lies in the fact that winter sports are intensively developing in this area, new ski resorts and infrastructure are being created.

Studies of snowiness and avalanche activity have been carried out in different mountainous regions: Iceland [1], the Alps [2–10], the Columbia and Rocky Mountains [11–15], the Caucasus [16–19]. In the Himalayas [20] and Tibet [21,22], only variability of snowiness was studied. All these areas, excluding the Himalayas and Tibet, belong to the areas

with a maritime or intermountain snow climate [23]. The Ile Alatau Ridge lies in the zone of a continental snow climate. Previously, research of variability of snowiness and avalanche activity was not carried out in the mountainous areas with this climate. There are very few publications about snow and avalanches on this region in English [24,25]. Therefore, this study could be relevant for understanding the features of a temporal variability of snowiness and avalanche activity in the mountainous regions with a continental snow climate.

Various indices can characterize winter snowiness and avalanche activity. The most commonly used indices of snowiness are the sum of winter precipitation and snow depth [4,5,12–15,22]. In [12], an index of avalanche activity is the avalanche size sum for a certain period. In [17], the maximum avalanche volume characterizes avalanche activity. In this work, the amount of winter precipitation and annual maximum snow depth were used to characterize winter snowiness. The total avalanche volume, maximum avalanche volume and number of avalanches were used as indices of avalanche activity.

## 2. Study Area

The Ile Alatau Ridge belongs to the mountain system of the Northern Tien Shan. It is located on the northern periphery of High Asia and stretches for 110 km along 43° N between 76° and 78° E. (Figure 1). The ridge rises above the Ile intermountain depression. The foot of the ridge lies at an altitude of about 1000 m above sea level. The main watershed is located at an altitude of more than 4000 m. The highest summit is the Talgar Peak with an altitude of 4976 m. The tree line of the Tien Shan spruce passes at an altitude of 2800 m. Glaciers lies above 3500 m.

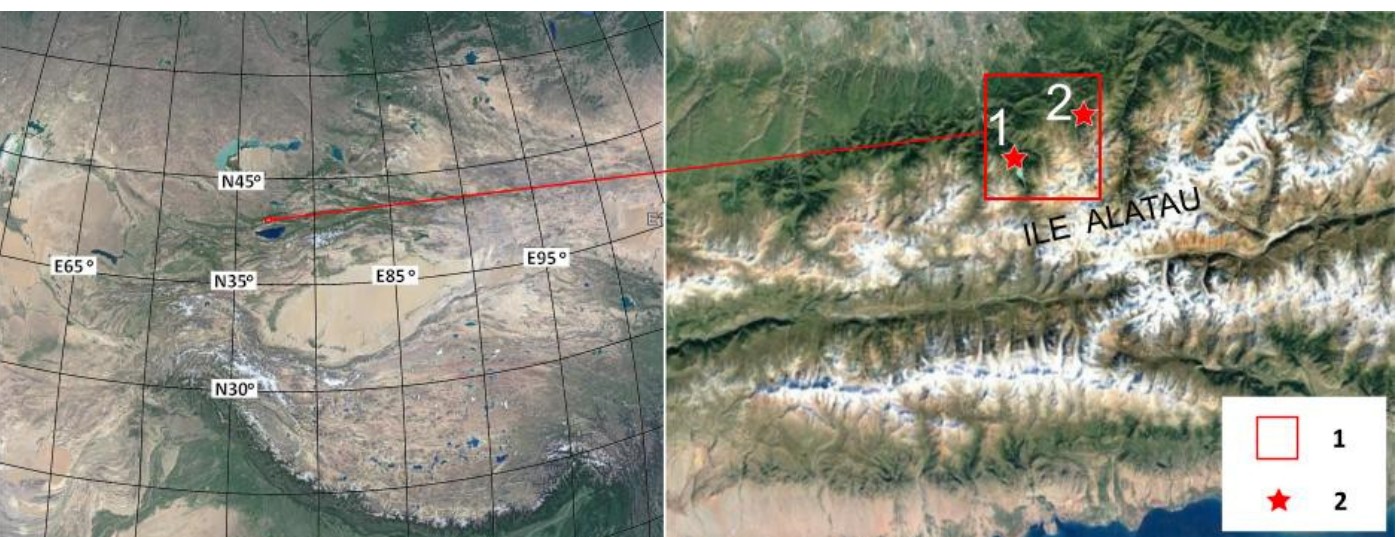

**Figure 1.** The location of the study area. **1**—Study area boundaries; **2**—Weather and avalanche observation stations: 1—Big Almaty Lake, 2—Shymbulak.

The study area occupies the middle parts of the Kishi and Ulken Almaty river valleys in the central part of the Ile Alatau. The study area is 150 km$^2$. The valleys have a U-shaped cross section profile. The excesses of the ridges above the bottoms of the valleys are 600–800 m; channeled avalanche paths of a complex structure predominate. The area of avalanche starting zones ranges from 1 to 50 ha.

The most favorable conditions for winter recreation are in the altitude zone from 2000 to 3200 m. In this zone, the snow cover, which is sufficient for skiing, remains from December to April. The maximum annual snow depth varies from 70 to 150 cm; the typical total height of a new snow is 100–300 cm. The highest avalanche hazard and avalanche activity are noted in the same altitudinal zone. An avalanche's dangerous period lasts from late November to an early April. The main causes of avalanches are snowfalls and thaws.

Weak layers in the snow cover usually form in late December or early January. At the beginning and in the middle of winter, dry snow avalanches caused by snowfalls prevail. At the end of winter, wet snow avalanches associated with thaws predominate. Medium-sized avalanches (1–10 thousand m$^3$) occur most often. About 10% of avalanches have a volume of more than 10 thousand m$^3$ and less than 1% of avalanches have a volume of more than 100 thousand m$^3$. The maximum-recorded avalanche volume was 350 thousand m$^3$ [25].

The Ile Alatau has a continental type of snow climate according to Armstrong's classification [23]. On the Big Almaty Lake meteorological station, located in the central part of the Ile Alatau at an altitude of 2516 m, for the period from 1936 to 2000, the average air temperature of a cold period was −6.8 °C, and the average maximum annual snow depth is 106 cm [26,27]. The average annual air temperature in the period from 1973 to 2014 increased at a rate of 0.0242 °C/year, and the annual precipitation amount increased at a rate of 0.58 mm/year [28]. The maximum daily precipitation is 60 mm. On average, during a winter there are 7 days with precipitation of more than 10 mm. Every two years there are winters when the daily precipitation exceeds 30 mm [27].

The depth of snow cover is highly dependent on the slope aspect. It is maximum on the northern slopes and two times less on the western and eastern slopes. A permanent snow cover is not formed on the southern slopes [24].

The common features of winters in the regions with a continental climate are low air temperatures and a shallow snow depth in November–February, as well as strong thaws and heavy snowfalls in March [25]. In January, the minimum air temperature may drop below −30 °C. At the same time, the snow depth usually does not reach 70 cm. As a result, a weak snow layer, composed by depth hoar, is formed in the lower part of the snowpack. Thickness of this layer can be up to 50 cm. This makes the snow cover unstable sometimes from the end of December. In March, there are often thaws up to 15 °C and snowfalls of 20–30 mm. Therefore, there is a peak of avalanche activity in March. This month, avalanches of maximum size occur, and the number of days with avalanches reaches 15–20 per month.

## 3. Materials and Methods

The work used the long-term data from two weather and avalanche observation stations: Big Almaty Lake (BAL) and Shymbulak (Figure 1). They belong to the State HydroMeteorological Service of Kazakhstan. Duration of the observation period is 57 years (1966–2022). The BAL station is located in the Ulken Almaty River valley at an altitude of 2516 m. The Shymbulak station is located in the Kishi Almaty River valley at an altitude of 2200 m, 12 km northeast of the BAL station. The observation points are representative for avalanche formation conditions for the central part of the Ile Alatau.

Weather observations have been made since 1936. Avalanche observations were started in 1966 to forecast avalanche danger in the basins of the Kishi and Ulken Almaty River valleys. Meteorological data was taken only for the period of avalanche observations. Observations of weather and avalanches were carried out according to the standard methodology of Kazakhstan [29,30], WMO [31] and Canadian Avalanche Association [32].

The database includes the annual value of the following variables: (1) winter precipitation, (2) snow depth, (3) total avalanche volume, (4) maximum avalanche volume, and (5) number of avalanches. Additional information on avalanche victims and damage was taken from the avalanche reports.

The sum precipitation from November to March was used as the winter precipitation. The average monthly air temperature is below 0 °C during these months, and precipitation mostly falls in the form of snow. Of course, the annual sum of a solid precipitation can be more accurately obtained by summing up the daily precipitation for a cold period, but such data are available only for the last 10 years. The data on monthly precipitation can be only obtained in the archives.

The average maximum annual snow depth on the northern slopes is used as the snow depth. The depth of snow on the northern slopes in the Ile Alatau conditions better reflects

the conditions of snow accumulation in avalanche starting zones, since snow remains better on the northern slopes than on the sites of the weather stations located at the bottoms of the valleys. The snow depth on the northern slopes is used to characterize winter snowiness together with winter precipitation, since there is no a strict correspondence between these characteristics.

The precipitation was measured by a Tretyakov's precipitation gage with a Nifer's windshield with an accuracy of 0.1 mm. The snow depth was measured by remote snow stakes located on the northern slopes near the starting zones of the reference avalanche paths. The observations were carried out with a frequency of 5–7 days. The measurements were carried out with the help of a spyglass from the bottom of the valley from a distance of up to 2 km. The measurement accuracy was 5 cm. The number of stakes varied from year to year from 15 to 20. The data from all the stakes was averaged. The maximum value of winter snow depth was chosen from them.

The number of avalanches and their volumes were determined in the reference avalanche paths. In total, 70 avalanche paths were under observation. 55 of them were channeled paths, and 15 were unconfined ones (Figure 2). The avalanche starting zones were located in the altitude range of 2000–3200 m. The area of starting zones varied from 1 to 50 ha.

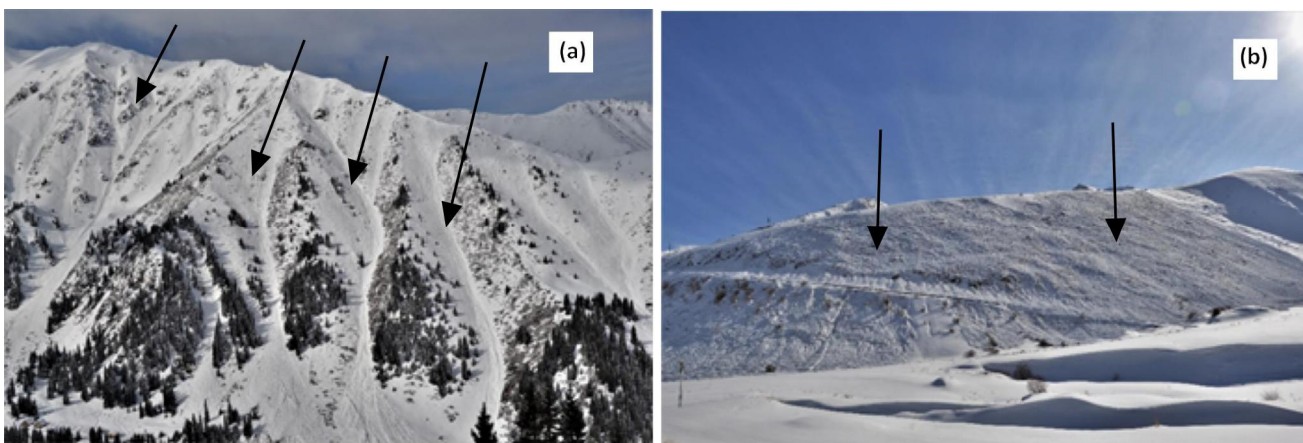

**Figure 2.** Avalanche paths (marked by arrows) in the Ulken Almaty valley: (**a**) channeled; (**b**) unconfined.

The volume of avalanches was determined from snow deposits. We've divided avalanche sizes into five categories according to the Canadian classification [32], used in Canada and the USA: (1) small, less than 0.1 thousand m$^3$, (2) medium, 0.1–1 thousand m$^3$, (3) large, 1–10 thousand m$^3$, (4) very large, 10–100 thousand m$^3$, and (5) extremely large, more than 100 thousand m$^3$. The volumes of avalanches larger than 10 thousand m$^3$ were measured using geodetic instruments and avalanche probes. The volumes of avalanches less than 10 thousand m$^3$ were estimated visually. When calculating the total avalanche volumes of small, medium and large avalanches were taken as 30 m$^3$, 300 m$^3$, and 3000 m$^3$ accordingly.

The maximum avalanche volume was defined as the volume of the largest single avalanche that occurred during a winter.

The number of avalanches was calculated by summing the number of all avalanches that occurred during a winter in the reference avalanche paths. Until 2000, only avalanches larger than 100 m$^3$ were registered at the avalanche stations. After 2000, the Observation Guide was amended [29], and observers began to note small avalanches, with a volume of less than 100 m$^3$. This led to increase in the number of avalanches, caused not by natural, but by human factors. The problem of subjectivity of the avalanche observation data, which complicates the analysis of the long-term data, was also noted in [4,11,13,14].

The following indicators of snowiness and avalanche activity variability were calculated: (1) standard deviation of long-term series (STD), (2) coefficients of the variation equal

to the ratio of STD to the long-term average value, (3) slopes of long-term temporal trends, (4) Spearman's correlation coefficients between variables, and (5) return periods of the maximum values of snowiness and avalanche activity indices.

The Spearman's correlation coefficient was chosen because distributions of the total avalanche volume and maximum avalanche volume have a significant positive skew.

The Gumbel distribution [33] was used to calculate a return period of the maximal snowiness and avalanche activity indices. It describes probability distribution of rare events that occur less than once in 10 years. The Gumbel distribution is recommended for the analyses of long-term series by WMO [34] and was used for estimation of the extreme snow loads in Kazakhstan [35].

The calculations of a return period were carried out in the following sequence: (1) the long-term series was rebuilt into a variational series in ascending order of values; (2) for each member of the variational series, the probability $p = (n - 0.5)/N$ was calculated, where $n$ is the number of the member of a variational series, $N$ is the number of members of the series, (4) the index $x = (-ln\,(-lnp))$ was calculated for each member; (4) the dependence of the variable values on $x$ was built; and (5) according to it the variable values were determined for various return periods $T$. In this case, the equation $p = (1 - 1/T)$ was used. Figure 3 shows an example of a graphical determination of return periods of winter precipitation.

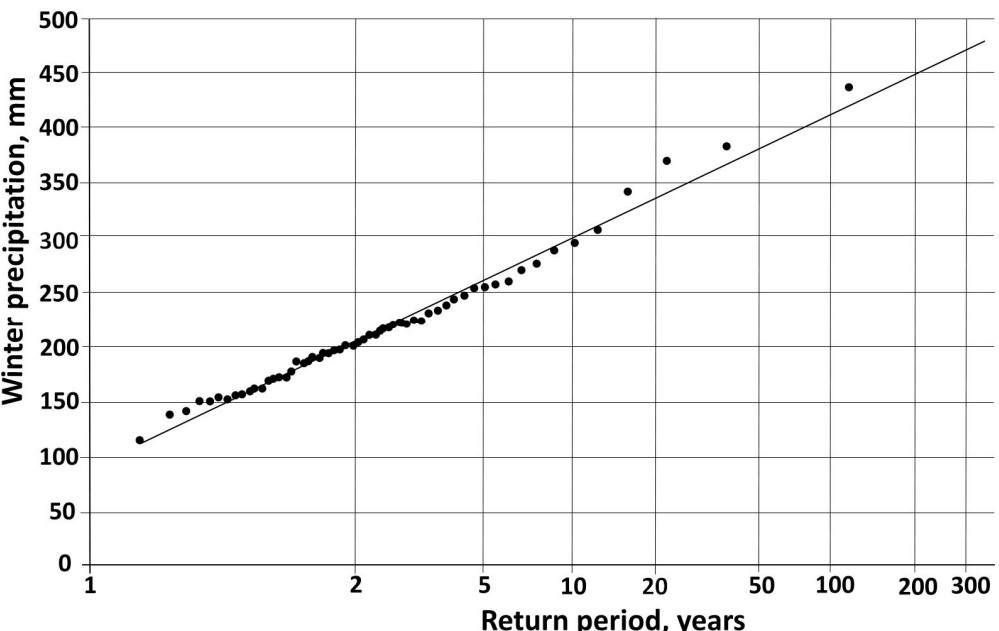

**Figure 3.** A plot of the winter precipitation with a different return period.

## 4. Results

### 4.1. Correlation between the Indices of Snowiness and Avalanche Activity

The Spearman's correlation coefficients between snowiness and avalanche activity indices are given in the Table 1. The correlation is considered very weak when a correlation coefficient is less than 0.2, weak when it is equal to 0.2–0.4, moderate when it is equal to 0.4–0.6, strong when it is equal to 0.6–0.8, and very strong when a correlation coefficient is more than 0.8.

The correlation between all indices of snowiness and avalanche activity is positive; that is, an increase in one index is accompanied by an increase in all other indices (Figure 4).

**Table 1.** The Spearman's correlation coefficients between snowiness and avalanche activity indices.

|  | Winter Precipitation | Snow Depth | Total Avalanche Volume | Maximum Avalanche Volume |
|---|---|---|---|---|
| Snow depth | 0.56 |  |  |  |
| Total avalanche volume | 0.54 | 0.53 |  |  |
| Maximum avalanche volume | 0.33 | 0.23 | 0.69 |  |
| Number of avalanches | 0.56 | 0.67 | 0.48 | 0.01 |

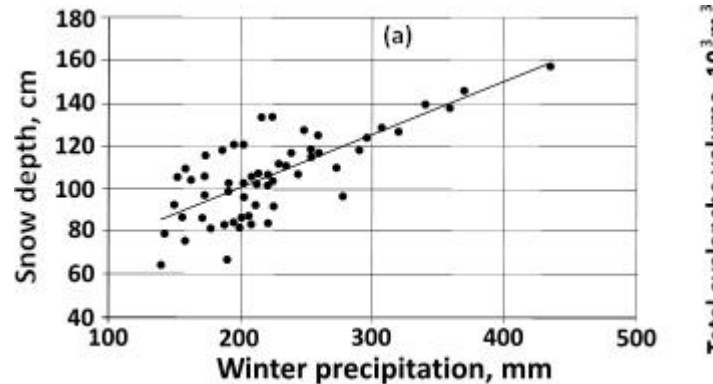 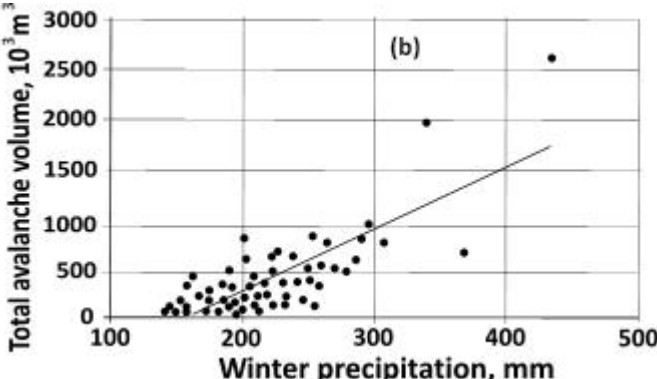

**Figure 4.** The correlation (**a**) between snow depth and winter precipitation; (**b**) between the total avalanche volume and the winter precipitation.

A moderate correlation is marked between snow depth and winter precipitation. The correlation between snowiness indices and the total avalanche volume is also moderate, but it is weak between them and the maximum avalanche volume. There is a moderate correlation between winter precipitation and the number of avalanches and a strong correlation between snow depth and that index. A strong correlation is determined between the total avalanche volume and maximum avalanche volume, and correlation is moderate between that index and the number of avalanches. The correlation between the maximum avalanche volume and the number of avalanches is very weak.

*4.2. Interannual Variability and Long-Term Temporal Trends of Snowiness and Avalanche Activity*

The characteristics of an interannual variability and long-term temporal trends of snowiness and avalanche activity are given in Figure 5 and Table 2. A rate of increase or decrease of a temporal change is expressed by the slope of a trend line.

Over the past 57 years, winter precipitation ranged from 116 mm (2004) to 436 mm (1966). The long-term average value was 218 mm, and the standard deviation was 62 mm, the coefficient of variation equal to 0.29 was moderate.

According to [17], winters are considered highly snowy when winter precipitation exceeds the long-term average value by more than a standard deviation, and winters are lowly snowy when winter precipitation is less than the average value minus STD. Thus, the winters of 1966, 1985, 1987, 1999, 2002, 2004, 2010, and 2017 are highly snowy, and the winters of 1968, 1974, 1978, 1982, 1983, 1989, 1996, 2005, and 2007 are lowly snowy in the Ile Alatau.

Over a long-term period, the winter precipitation shows a weak non-insignificant increasing trend with a rate of 0.37 mm/year. The rate has markedly increased in the last 20 years. During this period, it was 1.36 mm/year. The average value of winter precipitation for this period was 231 mm, compared with 209 mm for the period from 1966 to 2002.

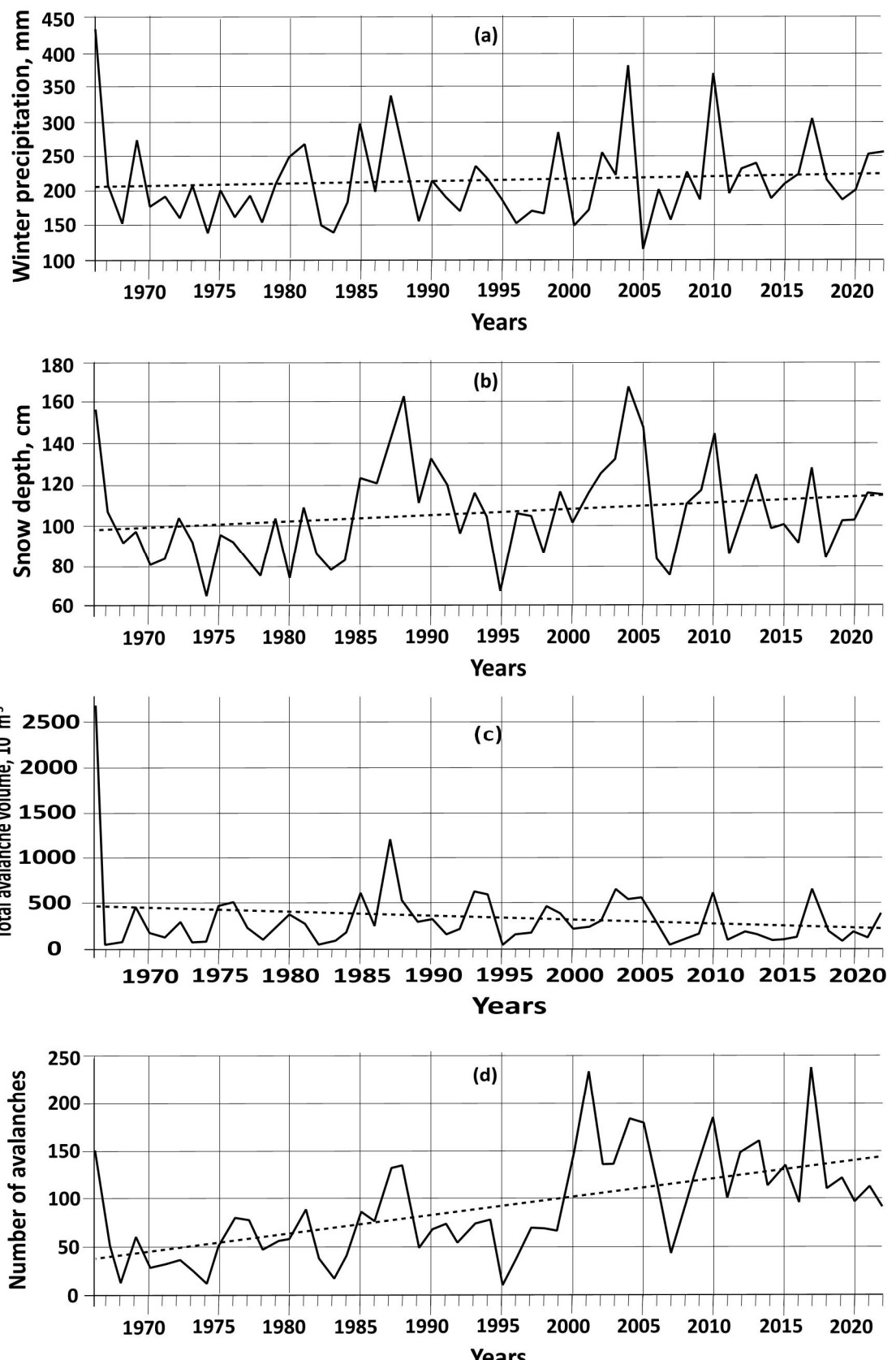

**Figure 5.** The interannual variability and the long-term temporal trends (dotted line) of snowiness and avalanche activity indices: (**a**) winter precipitation, (**b**) snow depth, (**c**) total avalanche volume, (**d**) number of avalanches.

**Table 2.** The characteristics of interannual variability and long-term temporal trends of snowiness and avalanche activity indices.

| Index | Long-Term Average Value | Standard Deviation | Variation Coefficient | Slope of Trend Line, Year$^{-1}$ |
|---|---|---|---|---|
| Winter precipitation, mm | 218 | 62 | 0.28 | 0.37 |
| Maximum snow depth, cm | 106 | 23 | 0.22 | 0.29 |
| Total avalanche volume, $10^3$ m$^3$ | 336 | 380 | 1.1 | −4.15 |
| Maximum avalanche volume, $10^3$ m$^3$ | 67 | 75 | 1.1 | −1.99 |
| Number of avalanches | 92 | 54 | 0.59 | 1.89 |

A long-term average value of the maximum snow depth was 106 cm, the maximum was 157 cm, the minimum was 68 cm, the standard deviation was 23 cm, and the variation coefficient was 0.22. Noting that a variation coefficient of snow depth was 0.32 in work [13] for the Kootenay Pass in Canada with the conditions of an intermountain snow climate.

The snow depth has a non-significant increasing temporal trend with a rate of 0.29 cm/year. The winters with a high snow depth were 1966, 1987, 1988, 2004, 2005, and 2010, the ones with a low snow depth were 1974, 1978, 1979, 1995, and 2007.

Distribution of the total avalanche volume is very variable (the coefficient of variation is equal to 1.2). It has a significant positive skew. The maximum of the total avalanche volume was recorded in 1966 (2590 thousand m$^3$), the minimum—in 1995 (35 thousand m$^3$). A non-significant decreasing temporal trend was found for this index. The decreasing rate is −4.15 thousand m$^3$ per year.

For 57 years, there were 12 years with a high avalanche activity, when the total avalanche volume exceeded 500 thousand m$^3$: 1966, 1976, 1985, 1987, 1988, 1993, 1994, 2003, 2004, 2005, 2010, and 2017. A very high avalanche activity was in 1966 and 1987, when the total avalanche volume exceeded 1 million m$^3$. A low avalanche activity, when the total avalanche volume was less than 100 thousand m$^3$, was observed in 1968, 1969, 1973, 1974, 1982, 1983, 1995, 2007, and 2015.

The maximum avalanche volume, alike the total avalanche volume, has a large variability (coefficient of variation is equal to 1.1) and a positive skew. The absolute maximum avalanche volume was recorded in 1966 (350 thousand m$^3$). The winters, when the maximum avalanche volumes exceeded 150 thousand m$^3$, were 1975, 1979, 1980, 1981, 1985, 1986, 1989, and 1994. The maximum avalanche volume was very small and less than 10 thousand m$^3$ in 1973, 1974, 1982, 1991, 2000, 2013, and 2015. A significant decreasing temporal trend with a rate of −1.99 thousand m$^3$ per year has been marked for the maximum avalanche volume. It has never reached 100 thousand m$^3$ over the past 23 years.

Over 57 years of observations, 5220 avalanches were registered in 70 reference avalanche paths. The maximum number of avalanches (241) was in 2017, the minimum (10)—in 1968. A distribution of the number of avalanches has moderate variability with the variation coefficient of 0.56 and a positive skew. The large number of avalanches was noted in 1966, 1987, 2001, 2004, 2005, 2009, 2010, 2013, and 2017, when it exceeded 150, the small one—in 1968, 1969, 1973, 1974, 1983, and 1995, when it was less than 30. The number of avalanches is increasing with a significant temporal trend of 1.9 avalanches per year. A sharp increase of avalanche number was noted after 2000. From 1966 to 1999, the average avalanche number was 71.6, and since 2000 it has risen to 131 avalanches per year.

*4.3. Return Periods of the Maximum Values of Snowiness and Avalanche Activity Indexes*

Table 3 shows the results of calculations of the maximum values of snowiness and avalanche activity indices with a return period of 10, 20, 50, 100, and 300 years. Gumbel's distribution was used for these calculations.

**Table 3.** The values of snowiness and avalanche activity indices with different return periods.

| Return Period, Years | Winter Precipitation, mm | Snow Depth, cm | Total Avalanche Volume, $10^3$ $m^3$ | Maximum Avalanche Volume, $10^3$ $m^3$ | Number of Avalanches |
|---|---|---|---|---|---|
| 10 | 298 | 137 | 928 | 176 | 154 |
| 20 | 341 | 152 | 1202 | 219 | 179 |
| 50 | 380 | 171 | 1652 | 273 | 212 |
| 100 | 410 | 185 | 1821 | 314 | 236 |
| 300 | 470 | 207 | 2240 | 379 | 275 |

### 4.4. Avalanche Accidents

From 1966 to 2022, there were 28 winters with avalanche accidents in the Ile Alatau (Figure 6). In total, 143 people were caught by avalanches during this time [36,37]. 67 of them were lost, 76 survived. Tourists, climbers and skiers account for 90 % of the victims. In most cases, people were captured by an avalanche caused by themselves. The volume of such avalanches did not exceed 10 thousand $m^3$. The most tragic accident happened in 1972 when an avalanche swept away 17 tourists, killing 8 of them.

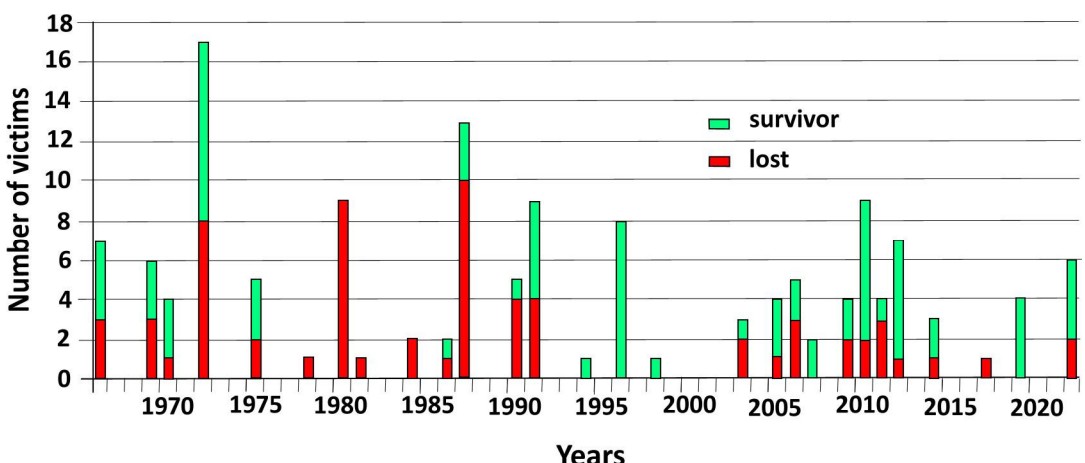

**Figure 6.** Avalanche accidents in the Ile Alatau from 1966 to 2022.

The years with a large number of victims, when six or more people were caught by avalanches, were 1966, 1969, 1972, 1980, 1987, 1991, 1996, 2010, 2012, and 2022. Most people were caught by avalanches in 1972, and most people were killed in 1987. Fluctuations in the annual number of avalanche victims are very irregular, but two time periods can be distinguished. The first one, with a relatively high number of victims, was before 2000. The second one, with a relatively low number of victims, was after 2000.

A significant material damage occurred in 1966, 1987, and 2010, when avalanches destroyed ski lifts, tourist camps and the forest over a large area.

## 5. Discussion

### 5.1. Correlation between Indices of Snowiness and Avalanche Activity

The analysis of correlations between winter snowiness indices (Table 1) showed an unexpected, at first glance, result. The correlation coefficient between winter precipitation and snow depth turned out to be only 0.56, although the relationship between these indices seemed obvious and should have been strong. It is explained by the fact that snow depth on slopes depends not only on the amount of precipitation, but also on their kind, in particular on fresh snow density and the rate of subsequent settling. In addition, some snow evaporates during the winter period. The processes of wind redistribution of snow also have their influence.

While winter precipitation and snow depth are increasing, there is an increase in all indices of avalanche activity, especially the total avalanche volume and the number of avalanches. At the same time, the maximum avalanche volume weakly correlates with winter snowiness, since it is strongly influenced by individual features of the avalanche path.

The total and maximum avalanche volumes are strongly related, since usually the maximum avalanche volumes in a given winter are observed in several avalanche paths. A moderate correlation between the total volume and number of avalanches is explained by the fact that small-sized avalanches predominate quantitatively, making a small contribution to the total avalanche volume. For the same reason, there is practically no correlation between the maximum avalanche volume and number of avalanches.

*5.2. Variability and Long-Term Trends of Snowiness and Avalanche Activity Indices*

A moderate variability of snowiness and significant variability of avalanche activity were determined in the Ile Alatau. This is in a good agreement with the results of studies in the North America and Europe mountains [2–5,11–14].

Statistically significant long-term temporal trends are determined for the maximum avalanche volume and number of avalanches. For the remaining indices of snowiness and avalanche activity, the trends are non-significant. The maximum avalanche volume is decreasing at a rate of $-1.99$ thousand $m^3$ per year, and avalanche number is increasing at a rate of 1.89 avalanches per year. However, if a decrease in the maximum avalanche volume occurs for natural reasons, then an increase in number of avalanches occurred for human reasons, due to the change in the method of avalanche registering and increasing avalanche control.

Higher winter precipitation years coincide with higher snow depth years in 43% of cases and with higher total avalanche volume years in 71% of cases. Large total avalanche volume years coincide with large avalanche number years in 50% of cases, and with large maximum avalanche volume years only in 25% of cases.

The comparison of avalanche winters in the Ile Alatau with avalanche winters in other mountainous areas showed that for the Caucasus they coincide in 40% of cases; this was in 1976, 1987, 1993, and 2010. The winters of 1987 and 2010 were especially severe. In the winter of 1987 in the Caucasus, winter precipitation and snow depth were 3–4 times higher than the long-term average values [17]. Avalanches occurred in the places they never usually did. 105 people died. This year was the second one peak of snowiness and avalanche activity in the Ile Alatau after 1966. Winter precipitation was 1.5 times higher than the average one. The total avalanche volume was 3.6 times higher, the maximum avalanche volume was 2.2 times higher, and the avalanche number was 1.5 times higher than usual. However, in the very highly avalanche active winters of 1969 in the Pamir [38] and 1999 in the Alps [39], in the Ile Alatau snowiness and avalanche activity differed little from the average ones. In the Canadian Columbia Mountains, the winter of 1972 was extremely highly avalanche active [12], but this winter was averagely avalanche active in the Ile Alatau.

A decrease in avalanche volume, both total and maximum, against the background of a slight, but still noticeable increase in winter precipitation and snow depth, over a long period remains not entirely clear, taking into account the presence of a positive correlation between snow cover and avalanche activity. This may be caused by several reasons.

Firstly, this is a change in the intra-annual distribution of winter precipitation. In recent years, a spring peak of precipitation has become more and more pronounced. At the same time, it shifts more and more to later dates. Therefore, in January and February, snow depth is less than the average long-term value and accounts for 50–100 cm. Under conditions of low air temperatures and high temperature gradients, a weak layer of 30–40 cm of thickness is formed in the snowpack. The snow cover becomes very unstable and even not very heavy snowfalls of 10–20 mm cause a large number of avalanches. Due to a shallow depth of snow, avalanche volume, as a rule, does not exceed 10 thousand $m^3$. As a result, in the

most active avalanche paths, a part of the snow is dumped from the starting zone to the bottom of the valleys, and very large avalanches are not formed there in spring.

Secondly, more and more avalanches occur as a result of active avalanche control. As a rule, artificial avalanches are smaller than natural ones. All of this leads to the fact that the proportion of smaller avalanches increases and frequency of very large avalanches decreases. This leads to a decrease in the total and maximum avalanche volumes. However, further research is required to better understand the causes of this phenomenon.

### 5.3. Return Periods of the Maximum Values of Snowiness and Avalanche Activity Indices

The recurrence of winters with an extreme winter precipitation, snow depth, volumes and numbers of avalanches is important for the organization of winter recreation, the cost of infrastructure maintaining and ensuring avalanche safety. Modeling of the frequency of winter snowiness and avalanche activity (Table 3) shows that once every 100 years, the amount of winter precipitation can be twice as much as the long-term average value. Snow depth exceeds the average value by 1.7 times with the same frequency. The maximum indices of avalanche activity deviate especially strongly from the average values. Once every 100 years, the total avalanche volume exceeds the long-term average by 5.4 times, the maximum avalanche volume exceeds it by 4.7 times, and the number of avalanches exceeds it by 3 times.

The maximum indices of snowiness and avalanche activity for 57 years of observations have a theoretical frequency much less than once in 57 years. Thus, the winter precipitation of 1966 can be repeated once in 120 years, the snow depth can be repeated once in 135 years, the total avalanche volume can be repeated once in 140 years, the maximum avalanche volume can be repeated once in 170 years, and the number of avalanches can be repeated once in 130 years.

### 5.4. Avalanche Accidents

Winters with a large number of victims coincide with highly avalanche active ones only in 30% of cases. People are often caught in self-initiated avalanches, and this can happen at a relatively low level of avalanche danger. A high material damage from avalanches was noted in 1966, 1987, and 2010. All these years were highly avalanche active ones.

The number of people caught in an avalanche depends little on natural avalanche activity. There are winters with low avalanche activity when there are many accidents; for example, the winters of 1972, 1980 and 1991 and, conversely, there are winters with high avalanche activity, when there is no avalanche accident; for example, the winters of 1976, 1985, 1988, 1993, and 2003. This is because most often people are captured by avalanches caused by themselves. Therefore, the number of victims does not depend on the level of avalanche activity, but on the level of recreational activity.

In contrast to the number of avalanche victims, material damage from avalanches is observed only in years with high avalanche activity, when avalanches descend where they rarely descend, or hit areas that they did not hit before.

### 6. Conclusions

Avalanche registration data is largely human dependent due to the changes in methods, observers, and their qualifications. In addition, natural avalanche activity is increasingly distorted by an active control. Therefore, in the studies of changes in avalanche activity, it is necessary to exclude the influence of these factors.

Since the indices of snowiness and avalanche activity have a significant or strong inter-annual variability and weak long-term temporal trends, when planning the development of winter recreation and economic development of the Ile Alatau territory, first of all, it is necessary to take into account the frequency of their extreme values. In this case, the temporal trends can be ignored.

Years of high avalanche activity may not coincide with years of high snowiness. Years with a high number of avalanche victims usually do not coincide with years with a high

avalanche activity, while years with a high material damage from avalanches always occur in such years. Therefore, warning the population about the level of avalanche danger and teaching tourists the rules of avalanche safety is of great importance to reduce the number of avalanche victims. Protective structures must be used to prevent material damage from avalanches.

According to the conditions of avalanche formation, the Tien Shan is an independent mountain region. Fluctuations in snowiness and avalanche activity are weakly associated here with the adjacent mountainous regions of the Pamir and Caucasus and are not associated with such remote areas as the Alps and North American mountains.

Variability of snowiness and avalanche activity is of a great scientific and practical importance. The research in this field must be continued in the little-studied mountainous regions of High Asia.

**Author Contributions:** Conceptualization, A.M. and V.B.; methodology, V.B. and V.Z.; formal analysis, A.M., V.B., V.Z., T.G. and S.R.; investigation, V.B., V.Z. and T.G.; data curation, V.Z.; writing—original draft preparation, V.B. and T.G.; writing—review and editing, V.B.; supervision, A.M. All authors have read and agreed to the published version of the manuscript.

**Funding:** This research was funded by the Science Committee of the Ministry of Science and Higher Education of the Republic of Kazakhstan, grant No AP09260155.

**Data Availability Statement:** The data presented in this study is available on request from the corresponding author.

**Acknowledgments:** The authors would like to thank the referees for useful comments and suggestions for improving the article and the staff of the Big Almaty Lake and Shymbulak Avalanche Stations for their help in collecting avalanche data and interpreting the results.

**Conflicts of Interest:** The authors declare no conflict of interest. The funders had no role in the design of the study; in collection, analyses, or interpretation of the data; in writing of the manuscript; or in the decision to publish the results.

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
