# Peer review of "Interannual Variability of Snowiness and Avalanche Activity in the Ile Alatau Ridge, Northern Tien Shan"

_water, doi:10.3390/w14182936_

Round 1

Reviewer 1 Report

The references in form [16/18] are strange to me. Should be either [16–18] or [16, 18]

Line 41 – [15–18], not [15–184]

Line 138 – “The Excel program was used…” is a bit childish and is not required. It does not matter if this was Excel or C++, besides, this is clear from the figures as presented (I would suggest to redraw them in some other graphics software)

Line 146 – “precipitation from November to March is more important than from November to April” is unclear and has no place in cited Table and Figure

Line 148 – Is not the correlation between winter precipitation and snow depth evident? And is not this SD i.e. “[annual] maximum snow depth on northern slopes”? The other correlations (and no correlation) are interesting results.

Table 2 – “Correlation” with what? Should be “Slope of linear trend”

Line 246 – “avaalanche” and “foe”

Figure 5 – understandable, but confusing. Basically, the talk is about probability curve with Gumbel distribution. Why not to make additional axis directly with probability/return period? Or one more Figure with probability? Right now, it is not clear how x in recalculated to return period for Table 3.

Line 260 – “avalnche”

“maximum volume of a single avalanche” – is not this “volume of a maximal avalanche in a year”? Was also in other paragraphs

Line 257 – “the increase in the number of avalanches occurred for subjective reasons…” should be related to correlation coefficients. Otherwise, what they are presented for?

Reviewer 2 Report

General comments

This paper analyzes seasonal trends in the amount of snowfall and avalanche activity in the Tien Shan mountains of Kazakhstan. The authors analyze a 57-year data set consisting of weather and avalanche observations from three stations, where snowiness is described by total winter precipitation and maximum snow depth and avalanche activity is described by estimated avalanche volumes and the number of avalanches. General patterns for the region are presented with correlations, timeseries, and extreme value statistics. The results provide a basic characterization of the snow climate and avalanche hazard for this area, which could be useful for planning and managing avalanche risk in the area.

While the work offers local value by better understanding the snow and avalanche climate, I think additional work is needed to emphasize the relevance to the scientific community. The dataset is relatively strong and well suited for this type of analysis and the overall research design is appropriate, however there are several issues in the structure of manuscript, presentation of the results, and most importantly an explanation of how the results support the objective of the study.

Please see my detailed comments below. While I think the article needs major revisions to provide a scientific contribution, and I would be happy to offer more feedback and further reviews.

Specific comments

·        In my opinion the manuscript could provide a more in-depth discussion of the relevance of the results. The introduction nicely frames the need for the study to inform tourism development in the region, however many of the concluding arguments report statistics rather than discuss their meaning. For example, the extreme value statistics reported in Table 3 seem highly relevant to the purpose of the study, but are minimally discussed. I think the implications of the temporal trends and the derived return periods could be further discussed to support the objective of the study.  

·        The methods and results are not particularly novel, as they cite numerous similar studies from other regions. In light of that, I think it is important for this paper to highlight the relevance of applying these methods in a study area that is less understood that previously studies areas such as Europe and North America. It should be clear to the reader why having this baseline understanding and climatology is important, especially in the conclusions and abstract

·        The description how the data was collected and analyzed could be improved. Certain details are missing and others are dispersed throughout various sections of the manuscript like the Introduction and Results (see Technical comments for examples). I suggest consolidating the Methods section to provide a more complete and comprehensive description of the study.

·        The snow water equivalent data is described in the Abstract, Introduction and Methods but is not presented anywhere in the Results or Conclusions. Either these results should be added and discussed or mention of SWE removed.

·        The overall presentation quality of the manuscript should be improved. There are numerous typos and grammatical errors, and several of the figures are not properly labelled or formatted. Also, the extensive use of abbreviations impacts the readability of the manuscript. Many of the abbreviated variables could be more easily understood with plain language (e.g., writing correlation coefficient rather than CC, and winter precipitation rather than WP). I recommend being more strategic with the use of abbreviations, perhaps only abbreviate the core snowiness and avalanche variables, and then use plain language for other terminology.

Technical comments

·        Line 17-19: It is not clear that “positive and negative trends” are temporal trends, it could be implied these trends refer to correlations. Also, more context is needed to know what is meant by amount of precipitation being the “most important” indicator.

·        Line 20: The abstract is missing a concluding statement about the relevant of the results and how they relate to the objective of the study.

·        Line 33: Stating the “only possible way” is too strong, as other methods may be possible (e.g., physical modelling, historic and cultural records, etc.)?

·        Line 39-47: While this summary of related literature provides a good background of how this study fits within existing work, a few details could be clearer. First, describing these studies as “quite rare” seems to be misleading when 21 citations are provided. Second, the Canadian studies were conducted in the Columbia Mountains not the Rocky Mountains. Third, I am not convinced none of these studies include a continental snow climate, as parts of the Alps and Caucasus ranges would be considered continental, although perhaps not as continental as the Himalayas and Tibet. This paragraph is where there can be greater emphasis on the need for basic snow and climate studies in the Tien Shan.

·        Line 53-59: It would be more conventional to define the variables analyzed in this study in the Methods section, and simply describe the types of indicators that have previously been used for snowiness and avalanche activity in the Introduction.

·        Line 73-83: It is unclear where these numbers come from. Please provide an explanation of how they were determined (and citation if possible). For example, saying “typical or average snow depths are …” would be clearer than definitive statements like “snow depths are..”.

·        Line 84-94: Describing the climate of this regions is helpful, however there should be more description of the data behind these climate figures. Is this from specific observation sites? Over what period? Is this for the entire Tien Shan or specifically the Ile Alatau? Are all of these values from the cited paper [23]?

·        Line 88-90: How are the beginning/middle/end of winter defined? Is stratifying maximum daily precipitation into these categories meaningful?

·        Fig. 1: Mark the location of the stations on the map would be helpful (e.g., MS, BAL, AS).

·        Line 69: Are these abbreviations necessary? They are rarely used elsewhere in the manuscript.

·        Line 105: How do the HMS standards for snow and weather observations compare / comply with international standards such as WMO or the International Classification for Seasonal Snow on the Ground (Fierz et al., 2009).

·        Line 124: Please provide more description of the avalanche observations such as who makes observations and why? Changes to the observation standards (i.e., pre and post 1999) should be introduced here rather than in the results.

·        Line 130: Please provide a better reference for avalanche size such as the Canadian Avalanche Association (2016) or the European standards https://www.avalanches.org/standards/avalanche-size/

·        Fig. 2: As someone unfamiliar with the region, I appreciate these photos because they nicely illustrate the terrain, snowpack, and avalanches in this study area. Thank you.

·        Line 138: It could be clearer that each indicator was aggregated into one value per season. The period over which WP was totalled should be mentioned here too. Based on the results it appears it was calculated over different periods, which should be described in the methods.

·        Line 141: Without additional description it is assumed Pearson correlation coefficients are calculated, however precipitation and avalanche data are left-bounded and right-skewed under which case a Spearman correlation is typically a better indicator of correlation.

·        Line 145-153: The descriptions of correlations here incorrectly use words such as “affect” and “effect”, which confuses the concepts of correlation and causation. Also, I assume the statement “most important” is intended to mean “strongest correlations”, which isn’t clear.

·        Line 146: The difference between the Nov-Mar and Nov-Apr datasets are not discussed anywhere.

·        Fig. 3/4: The linear regression equations should have the proper variable names rather than X and Y. Furthermore, neither the regression equations or r-squared values are described in the captions or discussed in the text. Are they necessary to report? The formatting of horizontal and vertical gridlines is also inconsistent.

·        Table 2, Fig. 3 and 4: Are correlation coefficients between the indicators and year meaningful? In the text the temporal trends seem to described with the regression slopes and their statistical significance rather than the correlation.

·        Line 190: Should it be STD instead of STL? This is an example of where plain language would be more readable than abbreviations.

·        Line 198 and 200: Are seasons with high/low SD and avalanche activity also determined by mean +/- one standard deviation? If so, it would be helpful to explicitly state how these winters were classified.

·        Line 230-236: Description of the Gumbel distribution, and derivation of return periods belongs in the Methods. I also recommend explaining why Gumbel distributions are an appropriate choice of extreme value distributions for snow and avalanche variables (e.g., Blanchett et al. , 2009; Katz et al., 2002) and then providing more summary and interpretation of the fit and return periods in Table 3.

·        Fig. 5: This figure is not adequately explained. It is showing how well WP fits the Gumbel distribution, but the x-axis variable is not described in the caption or axis label and the interpretation of this plot is not provided in the text. I would argue the return periods in Table 3 are the more important result to present.

·        Line 244: The definition and relevance of the “coefficient of reliability of approximation of dependencies of extreme values” has not been provided. What is this and why is it important?

·        Line 267-280: When discussing “heavy avalanche damage” and “fatalities” I wonder if there is any data available from this region about the impacts of avalanches (e.g., fatalities statistics, damage to infrastructure, etc.). Even if anecdotal, it would be helpful data to support the claims about the impacts of snowiness and avalanche activity in this region.

References

Blanchet, J., C. Marty, and M. Lehning. (2009). Extreme value statistics of snowfall in the Swiss Alpine region. Water Resource Resesrch, 45, W05424.

Canadian Avalanche Association (2016). Observation Guidelines and Recording Standards for Weather, Snowpack, and Avalanches, Canadian Avalanche Association, Revelstoke, BC, cdn.ymaws.com/www.avalancheassociation.ca/resource/resmgr/standards_docs/OGRS2016web.pdf.

Katz, R.W., M.B. Parlange, and P. Naveau. (2002). Statistics of extremes in hydrology. Advances in Water Resources, 25, 12871304

Fierz, C.R.L.A., Armstrong, R.L., Durand, Y., Etchevers, P., Greene, E., McClung, D.M., Nishimura, K., Satyawali, P.K. and Sokratov, S.A., 2009. The international classification for seasonal snow on the ground.

Round 2

Reviewer 2 Report

General comments

I appreciate the revisions made to the manuscript which now does a better job framing the relevance of the research and explaining the methods. In the technical comments below I list some specific places where I think additional revisions could be made to further improve the manuscript. Many of them are suggestions to improve grammar and readability. I have two remaining major concerns:

1.      The introduction of new data and methods in the Results section (where application of the Gumbel distribution is first described) and the Discussion section (where the avalanche accidents data is first mentioned and presented) which would be better fit in the Methods section.

2.      The use of terms such as “most important”, “depends”, “impacts”, and “affects” when describing results from the correlation analysis. There are several places where these terms imply causation when the analysis supporting the claim is simply from correlations. Examples are provided below.

Technical comments

·        Snow water equivalent is still included in the abstract, keywords, and introduction yet no SWE data or results are presented. This data should either be included in the analysis or description of SWE should not be mentioned so strongly in the beginning sections of the paper.

·        Line 18: It would be much clearer to add “insignificant temporal trend” here.

·        Line 19-21: The claim precipitation is the most important is strong when only supported by a correlation analysis, it would be better to explicitly state that precipitation had the strongest correlation with avalanche indicators.

·        Line 27: “…amount of snow is the main…”

·        Line 49: “Therefore, this study could be relevant to…”

·        Line 55: You could delete the sentence “In this work, we have used the most important of them.”, since this is explained in the next sentence. Also, I do not see the point of mentioning SWE in line 58 since it is not analyzed in this study.

·        Line 85: It sounds unusual to assert that weak layers always form in mid-December, because this would likely vary depending on specific weather patterns each season. I still think it could be clearer that lines 81-85 are describing trends rather than absolute values by adding words like “typical” or “average”.

·        Line 109: “unstable” is more clear than “weakly stable”

·        Line 117: “BAL station is located in the Ulken…”, same in next sentence.

·        Line 120: “slopes”

·        Line 126: “number of avalanches”, avalanche number sounds like an index describing each individual event

·        Line 129: “most” precipitation falls as snow

·        Line 136: “in avalanche starting zones”

·        Line 158: Thank you for better describing avalanche size classes. Please provide units for each volume (e.g., units after 0.1 - 1), as it is unclear whether these are in m3 or thousand m3.

·        Line 176: Please be more specific about “long-term linear trends’ (e.g., slope from a simple linear regression?)

·        Table 1: Why are the variables in the rows and columns in different orders, this makes it difficult to read.

·        Line 182: This argument about winter precipitation being the most important variable is questionable based on the correlations, as snow depth has comparable or even stronger correlations with some of the avalanche indicators. This paragraph remains somewhat confusing by providing a sentence for strong/moderate/weak correlations, and would be more logical to provide sentences for each concept such as how the snowiness variables correlate with each other, how the avalanche variables correlate with each other and finally how the snowiness variables correlate with the avalanche variables.

·        Line 209: “long-term average value

·        Line 213: Please specify that this is a positive or increasing trend.

·        Figure 3: “Relationship…” between” may be more appropriate than “Dependencies” in the caption as dependencies implies causation.

·        Figure 4: Just a thought that a barplot could be more appropriate than using a line to connect adjacent seasons since you don’t necessarily expect adjacent seasons to be related to each other.

·        Line 243: How is “very high” defined? Based on Fig. 4c 2010 does not seems much higher than some other high seasons such as 1993, 2003, and 2017, so I assume a specific threshold was used here.

·        Line 300: “phase of precipitation” instead of “mode of precipitation”?

·        Line 303: It is not clear what is meant by “increasing”, does this mean temporal trends or is it meant to somehow describe correlation.

·        Line 322-325: It could also be interesting to discuss that avalanche volume is decreasing despite precipitation and snow depth are increasing. Increasing snow depths don’t exactly align with the climate warming explanation for this region.

·        Line 341: Please define “avalanche winters”, does this refer to the same total avalanche volume threshold used to classify winters in the results section?
